# A Deep Learning Framework Integrating Multi-View Morphologic and Hemodynamic Features for Pericardial Disease Classification

**Sihyeon Jeong**[*1] (iD)                                      SIHYEON@YONSEI.AC.KR

**Jina Lee**[*2] (iD)                                             QQWWDJ@GMAIL.COM

**Yeonggul Jang**[†3] (iD)                                       YGJANG@ONTACTHEALTH.COM

**Jaeik Jeon**[3] (iD)                                           JAEIK.JEON@ONTACTHEALTH.COM

**Dawun Jeong** [1] (iD)                                       DAWUNJEONG96@YONSEI.AC.KR

**In Tae Moon** [4] (iD)                                          SMNOENEMY@GMAIL.COM

**Seung-Ah Lee**[3] (iD)                                      SEUNGAH.LEE@ONTACTHEALTH.COM

**Yeonyee E. Yoon** [3,5] (iD)                                   YEONYEEYOON@SNUBH.ORG

**Hyuk-Jae Chang**[2,3] (iD)                                          HJCHANG@YUHS.AC

[1] *Department of Internal Medicine, Graduate School of Medical Science, Brain Korea 21 Project, Yonsei University College of Medicine, Seoul, Republic of Korea*

[2] *Division of Cardiology, Severance Cardiovascular Hospital, Yonsei University College of Medicine, Yonsei University Health System, Seoul, Republic of Korea*

[3] *Ontact Health, Seoul, Republic of Korea*

[4] *Division of Cardiology, Department of Internal Medicine, Uijeongbu Eulji Medical Center, Eulji University College of Medicine, Uijeongbu, Republic of Korea*

*5 Cardiovascular Center and Division of Cardiology, Department of Internal Medicine, Seoul National University Bundang Hospital, Seongnam, Republic of Korea*

**Editors:** Under Review for MIDL 2025

## Abstract

Pericardial diseases require accurate and timely diagnosis, yet echocardiography analysis typically depends on expert interpretation. In this study, we introduce a novel two-stage deep learning framework designed to improve diagnostic accuracy by integrating multi-view echocardiographic video data— PLAX (parasternal long-axis), A4C (apical 4-chamber), modified A4C (modified apical 4-chamber), S4C (subcostal 4-chamber)— with hemodynamic indicators derived from IVC (inferior vena cava). In Stage 1, a tailored spatiotemporal convolutional neural network (CNN) effectively captures dynamic cardiac patterns, enabling precise classification of pericardial effusion severity and pericardial thickening (accuracy 0.921). In stage 2, embedding and integration of IVC-derived hemodynamic features substantially enhance sensitivity for detecting clinically significant cases (positive 0.969, negative 0.618). Our findings highlinght the clinical benefit of combining spatiotemporal echocardiographic features with functional indicators, potentially reducing reliance on subjective interpretation while ensuring compatibility with existing clinical workflows.

**Keywords:** echocardiography, deep learning, multi-view, fusion

---

[*] Contributed equally

[†] Corresponding author

## 1. Introduction

Pericardial diseases are common and encompass a wide range of conditions, from small, asymptomatic effusions to life-threatening cardiac tamponade. Some cases resolve spontaneously, while others progress to constrictive pericarditis with adhesion and thickening. Accurate diagnosis and treatment rely on understanding this complex pathophysiology and evaluating both structural and functional changes (Asteggiano et al., 2015). However, analyzing echocardiographic images of pericardial disease remains challenging, due to its complex nature requiring specialized skills in both image acquisition and interpretation. (Pepi and Muratori, 2006). Although recent deep learning approaches have attempted to automate echocardiographic analysis (Cheng et al., 2023; Chiu et al., 2024; Holste et al., 2024), most rely on single-view inputs or simply combine independent single-view models. These approaches are limited in capturing the detailed structural information and temporal dynamics of cardiac function. To address these limitations, we propose a multi-view framework that incorporates hemodynamic indicators derived from the inferior vena cava (IVC) for more precise assessment of pericardial disease. Indicators such as IVC dilatation and plethora, which are not easily visible in standard echocardiographic views, provide critical insights into elevated intrapericardial pressure and impaired venous return (Himelman et al., 1988). By integrating these hemodynamic features with video-based analysis, our framework offers a comprehensive assessment of pericardial diseases while potentially reducing reliance on manual interpretation.

## 2. Method

**Dataset**: We utilized the AI-Hub Open AI Dataset Project (AI-hub, 2021), comprising 2,118 patients: 1,041 normal, 739 pericardial effusion, 139 cardiac tamponade, 101 constrictive pericarditis, and 98 effusive constrictive pericarditis. It was split into training, validation, and test sets (8:1:1 ratio by patient). Four standard B-mode echocardiographic views (parasternal long-axis, apical 4-chamber, modified apical 4-chamber, subcostal 4-chamber) were extracted, resampled to 224×224 pixels, and augmented via random cropping, rotation, flipping, and contrast adjustments. Additionally, we manually measured IVC diameter with respiratory variation from IVC view, labeling "dilation" (positive when $diameter_{max} \geq 21mm$) and "plethora" positive when $\Delta diameter < 50\%$) (Klein et al., 2013).

**Proposed method** : We propose a two-stage framework that initially extracts spatiotemporal features from multi-view echocardiographic videos to classify pericardial effusion severity and thickening, and then incorporates IVC measurements to assess hemodynamic significance. This structured approach enables the model to systematically learn morphological and functional information.

**Stage 1 : Multi-view spatiotemporal feature fusion** Spatiotemporal features are extracted from the four echocardiographic views using a modified R(2+1)D network. To facilitate robust feature representation, we fully fine-tune a pretrained model (Park et al., 2025) for each available view. Extracted features are fused at the patient level and passed through two classification heads: one for pericardial effusion severity (4-class:≤Trivial, Small, Moderate, Large) and the other for pericardial thickening or adhesion (binary). To effectively balance these parallel tasks, we employ an uncertainty-based multi-task loss (Kendall et al., 2018), which dynamically adjusts each task's contribution based on estimated difficulty.

**Stage 2: Embedding-based hemodynamic assessment** Leveraging binary labels for IVC dilation and plethora, we map these indicators into an embedding space through a fully connected layer. This embedding is concatenated with the fused multi-view video features along the feature axis to form a unified representation, which the final classifier uses to predict hemodynamic significance (negative or positive). To address class imbalance and emphasize clinically significant positives, we employ a focal loss function (Lin et al., 2017).

## 3. Results and Conclusion

Table 1: Performance comparison of morphological classification metrics

| | Pericardial effusion amount | | | | | | | | Pericardial Thickening or Adhesion | | | |
| | Normal/Trivial | | Small | | Moderate | | Large | | Negative | | Positive | |
| | PanEcho | Proposed | PanEcho | Proposed | PanEcho | Proposed | PanEcho | Proposed | PanEcho | Proposed | PanEcho | Proposed |
|---|---|---|---|---|---|---|---|---|---|---|---|---|
| Accuracy | 0.862 | **0.933** | 0.831 | **0.900** | 0.800 | **0.899** | 0.947 | **0.949** | 0.840 | **0.919** | 0.840 | **0.919** |
| Precision | 0.824 | **0.919** | 0.400 | **0.581** | 0.813 | **0.830** | **0.679** | 0.676 | 0.912 | **0.927** | 0.424 | **0.810** |
| Sensitivity | **0.941** | 0.958 | **0.621** | 0.562 | 0.236 | **0.747** | 0.864 | **0.955** | 0.902 | **0.979** | 0.452 | **0.617** |
| Specificity | 0.774 | **0.906** | 0.862 | **0.939** | **0.982** | 0.943 | **0.956** | 0.947 | 0.452 | **0.617** | 0.902 | **0.979** |
| F1-score | 0.878 | **0.915** | 0.486 | **0.651** | 0.366 | **0.783** | 0.760 | **0.779** | 0.907 | **0.952** | 0.438 | **0.638** |

Table 1 compares the morphological feature classification performance between PanEcho (Holste et al., 2024) and our proposed multi-view model. PanEcho is a state-of-the-art model for echocardiography interpretation, capable of addressing multiple clinical tasks using a view-agnostic ResNet3D backbone, which we fine-tuned on our dataset. Despite its generality, our approach achieves consistently higher accuracy, precision, and sensitivity across all categories. These results highlight the effectiveness of fusing spatiotemporal information from multiple echocardiographic views, enabling more robust detection of effusion and thickening. Table 2 further demonstrates the additional benefit of incorporating IVC-derived functional indicators for classifying hemodynamic significance. By embedding binary flags indicating IVC dilation and plethora, our method outperforms both PanEcho and a variant of our model without IVC flags integration. Notably, sensitivity for detecting clinically significant positive cases rises from 0.233 (without IVC flags) to 0.618 (with IVC flags), underscoring the importance of combining morphological and functional cues for diagnosing pericardial diseases. Additionally, our lightweight architecture also supports efficient multi-view fusion, simplifying training and accelerating inference.

Table 2: Impact of IVC-feature integration on diagnostic sensitivity and accuracy

| | Negative | | | Positive | | |
| | PanEcho | w/o IVC flag | Proposed | PanEcho | w/o IVC flag | Proposed |
|---|---|---|---|---|---|---|
| Accuracy | 0.804 | 0.843 | **0.916** | 0.805 | 0.843 | **0.916** |
| Precision | 0.899 | 0.907 | **0.934** | 0.324 | 0.259 | **0.778** |
| Sensitivity | 0.871 | 0.918 | **0.969** | 0.387 | 0.233 | **0.618** |
| Specificity | 0.387 | 0.233 | **0.618** | 0.871 | 0.918 | **0.969** |
| F1-score | 0.885 | 0.912 | **0.951** | 0.353 | 0.246 | **0.689** |

In conclusion, this study demonstrates the effectiveness of the proposed framework in diagnosing pericardial diseases by integrating multi-view echocardiography. The framework captures spatiotemporal dynamics from multi-view echocardiographic videos and leverages clinically relevant hemodynamic indicators, significantly improving diagnostic accuracy. Future work will include automating the IVC measurement process to minimize manual intervention and advancing real-time decision support capabilities in pericardial disease.

## Acknowledgments

This research was supported by a grant of the Korea Health Technology R&D Project through the Korea Health Industry Development Institute (KHIDI), funded by the Ministry of Health & Welfare, Republic of Korea (grant number : RS-2024-00332481) and Seoul R&BD Program(BT230080) through the Seoul Business Agency(SBA) funded by The Seoul Metropolitan Government.

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
