# OpenReview forum: "A Deep Learning Framework Integrating Multi-View Morphologic and Hemodynamic Features for Pericardial Disease Classification"
_MIDL.io/2025/Short_Papers — MIDL 2025 - Short Papers_

### Official Review · Reviewer_RDfc · 2025-04-28

**Rating:** 4
**Confidence:** 5

**Summary:**

This study focus on the automatic classification of pericardial disease from echocardiographic image sequences and hemodynamic information. The method evolves two classification steps, one to learn useful spatiotemporal features from multi-view and one to add hemodynamic features. The method has been evaluated on a dataset of 2,118 patients and compared with a state-of-the art method. Results promote the interest of integrating hemodynamic features to features extracted from echocardiographic image sequences.

**Strengths:**

The strengths of this work are:
1) the relevance of the topic of this study, any improvement/innovation of which could have a major impact in our field
2) the relevance of the experiments and the metrics that have been chosen.
3) the size of the dataset used for the experiments (i.e. 2,118 patients)
4) the idea of combining video and hemodynamic features

**Weaknesses:**

The main weaknesses of this article concern:
- the lack of information regarding the method that has been developed. Even if this is a short paper, the authors should have provided more information on the key aspect of their method, such as the architecture that have been used, the task that have been implemented to fine-tune their pre-trained model, the uncertainty-based multi-task loss that have been used.
- The lack of motivation to implement their two-stage strategy instead of optimizing all feature sources together.
- It is not clear why the authors chose to use two separate heads, one for each disease. Why not use a single head with multi-class prediction ?
- even if the authors have tested their model with/without the hemodynamic features, they could have performed a more complete ablation study taking into account the different part of their model.

---

### Decision · Program_Chairs · 2025-05-01

Accept